# Scorpion Venom Heat-Resistant Synthetic Peptide Alleviates Neuronal Necroptosis in Alzheimer’s Disease Model by Regulating Lnc Gm6410 Under PM_2.5_ Exposure

**DOI:** 10.3390/ijms26094372

**Published:** 2025-05-04

**Authors:** Chuhao Qin, Dongsheng Li, Jiahui Zhang, Ze Yin, Fasheng Li

**Affiliations:** College of Medical Laboratory, Dalian Medical University, Dalian 116044, China; 18141127366@163.com (C.Q.); 17640569845@163.com (D.L.); 15149229442@163.com (J.Z.); 15949422305@163.com (Z.Y.)

**Keywords:** PM_2.5_, Alzheimer’s disease, Lnc Gm16410, necroptosis, scorpion venom heat-resistant synthetic peptide

## Abstract

Recent epidemiological studies have indicated that exposure to particulate matter with an aerodynamic diameter of 2.5 μm or less in the ambient air (PM_2.5_) is significantly associated with an elevated risk of developing Alzheimer’s disease (AD) and its progression. Scorpion venom heat-resistant synthetic peptide (SVHRSP) exhibits anti-inflammatory and neuroprotective properties. However, the exact ways in which SVHRSP mitigates the progression of AD induced by PM_2.5_ are still unknown. Long non-coding RNA (lncRNA) plays a crucial role in various biological processes. Necroptosis, a form of programmed cell death, has garnered considerable attention in recent years. This study aims to investigate whether Lnc Gm16410 and neuronal necroptosis are involved in PM_2.5_-exacerbated AD progression and the mechanisms of SVHRSP in alleviating this process. Through the establishment of a PM_2.5_ exposure model in AD mice and an in vitro model, it was found that PM_2.5_ exposure could promote necroptosis and the down-regulation of Lnc Gm16410, thereby promoting the progression of AD. Behavioral tests showed that SVHRSP alleviated cognitive impairment in PM_2.5_-induced AD mice. WB, qRT-PCR, and other molecular biological assays indicate that Lnc Gm16410 regulates neuronal necroptosis under PM_2.5_ exposure via the p38 MAPK pathway. SVHRSP is a potential regulator of AD progression by regulating Lnc Gm16410 to alleviate PM_2.5_ exposure-induced necroptosis. These findings offer new insights into the mechanism through which PM_2.5_ exposure accelerates the progression of AD, examined from the perspective of LncRNA. Furthermore, we offer new targets for the treatment and prevention of AD following PM_2.5_ exposure by investigating the mechanism of action of SVHRSP in alleviating AD.

## 1. Introduction

In recent years, ambient air pollution has emerged as a prominent cause of non-communicable diseases globally [1]. Particulate matter (PM), regarded as the primary pollutant in air pollution, is categorized into PM_10_, PM_2.5_, and PM_0.1_ according to their aerodynamic diameter. Among them, fine particulate matter (PM_2.5_), with an aerodynamic diameter of 2.5 μm or less, is one of the most harmful environmental risk factors [2]. PM_2.5_ consists of solid and liquid particles from both natural and anthropogenic sources; its primary toxic components include heavy metal particles, polycyclic aromatic hydrocarbon organic pollutants, nitrogen oxides, and some biological pollutants [3]. Environmental PM_2.5_ pollution is estimated to cause approximately 4.2 to 8.9 million premature deaths worldwide [4]. Due to its long suspension time in the air and small particle size, it is easy to be inhaled into the deep respiratory tract, which can lead to many respiratory diseases such as pneumonia and pulmonary fibrosis [5,6]; it can also enter the alveoli and bronchioles, causing serious harm to cardiovascular, nervous, and other systems with blood circulation [7,8].

Alzheimer’s disease (AD) is a neurodegenerative disease characterized by extracellular plaques containing beta-amyloid protein (Aβ) and intracellular neurofibrillary tangles (NFTs) containing tau. AD usually presents as significant amnestic cognitive impairment, but, less commonly, it also presents as non-amnestic cognitive impairment [9]. It is currently estimated that more than 55 million people worldwide are living with dementia, and the number of people affected will more than double by 2050 [10,11]. The mechanisms underlying the development and progression of AD are complex. In the ApoE cascade hypothesis, differences in genes such as ApoE2 and ApoE4ε4 can affect cell homeostasis, including cellular stress and lipid metabolism abnormalities, which, in turn, may lead to changes in the AD phenotype, such as neuroinflammation, vascular dysfunction, and synaptic loss [12]. P-tau217 is associated with brain atrophy and cognitive impairment in AD patients, and immunotherapy targeting p-tau217 improves tau disease in mice [13]. In addition, a large amount of Aβ deposition can also drive tau pathology and neuron loss through the p38 MAPK-MK2 axis [14]. Studies on animal models and humans have shown that environmental AD risk factors, such as diet, lifestyle, alcohol, smoking, and pollutants, can induce epigenetic modifications of key AD-related genes and pathways [15]. A long-term study of more than 63 million older Americans showed that for every 5 μg/m^3^ increase in annual PM_2.5_ concentration, the risk of first hospitalization for AD increased by 13%, and this risk continued to rise even after exposure to safe levels of PM_2.5_ [16]. It can be seen that PM_2.5_ exposure plays an important role in the occurrence and progression of AD. Despite the current development of anti-Aβ immunotherapy, tau-targeting drugs, and other treatments for AD [17,18], these methods have proven less effective in practical applications. Studies have shown that anti-Aβ immunotherapy to reduce plaque load often leads to a higher incidence of whole brain volume loss or ventricular enlargement in treated individuals [19]. Therefore, the current research on mature treatment and prevention methods of AD is of great significance.

Necroptosis is a novel form of programmed cell death characterized by the loss of membrane integrity and the release of intracellular contents [20]. Under the influence of death-inducing cytokines, RIPK1, a receptor-interacting serine/threonine protein kinase, interacts with the RHIM of RIPK3, leading to the phosphorylation and subsequent activation of RIPK3. The Ser345 residue in mice or the Thr357/Ser358 residues in humans, located within the mixed lineage kinase domain-like protein (MLKL), are recruited and phosphorylated, triggering oligomerization and insertion into the plasma membrane, leading to cell membrane integrity destruction and necroptosis [21,22]. Necroptosis has been shown to be an important pathway in many pathologies, such as the loss of HIF1α, which enhances IEC necroptosis, triggers intestinal inflammation, and exacerbates arthritis [23]. By reprogramming the bone marrow microenvironment via the release of IL-1α, necroptosis enhances the immune-dependent response of breast tumors to chemotherapy [24]. SFTPA1 promotes increased necroptosis of type II alveolar epithelial cells through the IRE1α-JNK axis and the progression of idiopathic pulmonary fibrosis (IPF) [25]. Although relevant studies have shown that PM_2.5_ exposure can cause a variety of cell death modes, such as ferroptosis [26], autophagy [27], and pyroptosis [28], there are few studies on the mechanisms related to PM_2.5_ exposure and necroptosis.

With the development of high-throughput sequencing technology, non-coding RNAs have received extensive attention. LncRNAs are generally considered to be a class of non-coding RNAs longer than 200 nucleotides (200 nt), but there is also support for defining them as non-coding transcripts longer than 500 nt [29]. LncRNAs can play an important role in many life activities through RNA–RNA, RNA–DNA, and RNA–protein interactions and act as a bridge to recruit related complexes [30,31]. Studies have shown that LncRNA NEAT1 directly binds and acts as a scaffold for PGK1/PGAM1/ENO1 complex assembly, promoting the glycolysis, proliferation, and metastasis of breast cancer [32]. The transcription factor DRG ELF1 binds to the LncRNA NIS promoter, aggravating neuropathic pain and notional hypersensitivity during maintenance [33]. LncRNA CARDINAL acts as a translation regulator that interacts with DRG1 and blocks its interaction with DFRP1, thereby regulating the rate of protein translation in the heart in response to stress [34]. Lnc Gm16410 is a long-chain non-coding RNA that is highly correlated with PM_2.5_ exposure, which was screened by high-throughput sequencing technology in previous laboratory studies. Studies have shown that Lnc Gm16410 is involved in PM_2.5_-induced endothelium–mesenchymal transformation and macrophage activation [35,36]. However, the specific mechanism of Lnc Gm16410 in PM_2.5_-mediated AD progression remains to be investigated.

Scorpions have been utilized in Chinese medicine for more than 1000 years and are extensively employed in traditional Chinese medicine to treat various ailments, including epilepsy, convulsions, and rheumatism. Scorpion venom (SV), a principal active component of scorpions, has surfaced as a significant therapeutic agent. An advanced understanding of its structural and functional properties, coupled with a growing body of evidence, indicates that components within scorpion venom, such as scorpion venom peptides, have significant potential for the treatment of various diseases, including cancer, epilepsy and infectious diseases like hepatitis B and C [37,38]. Scorpion venom heat-resistant peptide (SVHRP) is a type of active polypeptide extracted from scorpion venom and is analyzed for its amino acid sequence using LC-MS. Scorpion venom heat-resistant synthetic peptide (SVHRSP) is an active polypeptide synthesized according to its amino acid sequence. The substance exhibits properties of low toxicity, high purity, and heat stability, which are advantageous for its application in medical research [39]. Previous studies have shown that SVHRSP protects dopaminergic neurons by blocking p47^phox^ membrane translocation, inhibiting NOX2 activation and weakening microglia activation and M1 polarization [40]. In addition, SVHRSP may ameliorate age-related cognitive deficits by inhibiting oxidative stress and neuroinflammation [41]. SVHRSP, with its significant role in anti-inflammatory and nerve conservation, emerges as a promising candidate for innovative treatment in neurodegenerative diseases, echoing the potential of stem cell therapy as a new frontier in addressing these conditions.

This study aims to elucidate the regulatory mechanisms of necroptosis induced by Lnc Gm16410 in PM_2.5_-mediated AD progression and investigate the repair effects of SVHRSP. The findings will offer a novel theoretical and experimental foundation for early the diagnosis, treatment, and prognosis of AD in the context of PM_2.5_ exposure.

## 2. Results

### 2.1. PM_2.5_ Component Analysis

The primary components of PM_2.5_ include organic carbon, elemental carbon, polycyclic aromatic compounds such as fluoranthrene, chrysene, and benzo [a] pyrene. Additionally, water-soluble ions, like NH_4_^+^, Na^+^, K^+^, SO_4_^2-^, and NO_3_^-^, and metal elements, including Cd, Al, and Ag, were present. Under transmission electron microscopy, these particles typically cluster together, forming dense aggregates. Detailed results are in previous studies [42].

### 2.2. PM_2.5_ Exposure Promoted the Necroptosis of AD Neuron Cells

PM_2.5_ exposure was associated with the progression of AD [43]. Initially, the PM_2.5_ exposure model of the AD cell line was established, and two neuronal cell lines, HT22 and SH-SY5Y, were chosen. Utilizing the CCK8 assay, it was observed that the survival rate of both neuron cell types diminished progressively with escalating concentrations of Aβ_25-35_ exposure. Considering the cellular damage inflicted by co-exposure, 15 μM and 20 μM were selected as therapeutic concentrations for HT22 and SH-SY5Y cells, respectively. Subsequently, it was determined that the IC50 values for HT22 and SH-SY5Y cells under PM_2.5_ exposure were 174.1 and 233.1 μg/mL, respectively, following co-exposure to Aβ_25-35_ and PM_2.5_ (Figure 1A,B). In this study, to investigate the effects of PM_2.5_ exposure on the occurrence and progression of necroptosis in AD neurons, we detected necroptosis-related markers, such as RIPK1, p-RIPK1, MLKL, and p-MLKL, in HT22 and SH-SY5Y cells. The results of Western blot experiments indicated that in both HT22 and SH-SY5Y cells, the levels of RIPK1 and MLKL protein phosphorylation were elevated in the Aβ_25-35_ and Aβ_25-35_+PM_2.5_ groups compared to the control group. The phosphorylation levels of RIPK1 and MLKL in the Aβ_25-35_+PM_2.5_ group increased progressively with the extension of exposure time, with the most pronounced changes observed at the 48 h mark. The phosphorylation levels of RIPK1 and MLKL in the Aβ_25-35_+PM_2.5_ 48 h group were higher than those in the Aβ_25-35_ group (Figure 1C,D). In order to explore the impact of PM_2.5_ exposure on the occurrence and progression of AD, a Western blot assay was employed to monitor alterations in AD-related markers within HT22 and SH-SY5Y cells. The exposure duration for the Aβ_25-35_ and PM_2.5_ group was set at 48 h, while the Aβ_25-35_ and PM_2.5_ co-exposure group was subjected to 24 h, 36 h, and 48 h of treatment. Our findings revealed that the expression levels of IL-1β and TNF-α in HT22 and SH-SY5Y cells were elevated in the Aβ_25-35_ groups relative to the control group. The expression levels of IL-1β and TNF-α proteins in the Aβ_25-35_+PM_2.5_ 48 h group were higher than those in the Aβ_25-35_ group (Figure 1E,F). A similar trend was also observed in the qRT-PCR experiment (Figure 1G). These results indicated that PM_2.5_ exposure promoted the necroptosis of AD neuronal cells in a time-dependent manner.

### 2.3. Lnc Gm16410 Plays a Role in the Regulation of Neuronal Necroptosis in AD Following Exposure to PM_2.5_

To investigate whether Lnc Gm16410 plays a role in PM_2.5_-mediated AD progression, a qRT-PCR assay was utilized to detect the levels of Lnc Gm16410 in HT22 cells exposed to PM_2.5_. We found that the mRNA expression of Lnc Gm16410 was reduced in both the Aβ_25-35_ and Aβ_25-35_+PM_2.5_ groups compared to the control group. The mRNA expression of Lnc Gm16410 in the Aβ_25-35_+PM_2.5_ group decreased gradually with the extension of exposure time, with the most significant change observed at 48 h. The mRNA expression of Lnc Gm16410 in the Aβ_25-35_+PM_2.5_ 48 h group was lower compared to the Aβ_25-35_ group (Figure 2A). To further investigate the role of Lnc Gm16410 in the progression of AD under PM_2.5_ exposure, we constructed an overexpressed plasmid for Lnc Gm16410, which was transiently transfected into HT22 cells. The qRT-PCR results confirmed that the plasmid effectively up-regulated the expression of Lnc Gm16410 (Figure 2B). Subsequently, we explored whether Lnc Gm16410 was involved in the regulation of necroptosis during AD progression.

Cells from the control group and the OE-Lnc Gm16410 group were exposed to Aβ_25-35_ and PM_2.5_. The Western blot results indicated that in HT22 cells, the protein phosphorylation levels of RIPK1 and MLKL in the Aβ_25-35_+PM_2.5_ group were increased compared to those in the Aβ_25-35_ group, while the protein phosphorylation levels of RIPK1 and MLKL were significantly decreased following the overexpression of Lnc Gm16410 (Figure 2C). Subsequently, a Western blot assay was conducted to detect changes in AD-related markers in HT22 cells. It was observed that the expression of IL-1β and TNF-α protein in the Aβ_25-35_+PM_2.5_ group was elevated compared to the Aβ_25-35_ group. However, after the overexpression of Lnc Gm16410, the expression of IL-1β and TNF-α protein was significantly diminished (Figure 2D). The qRT-PCR experiment also exhibited the same trend. (Figure 2E). These findings indicate that Lnc Gm16410 plays a role in the regulation of neuronal necroptosis in AD under PM_2.5_ exposure.

### 2.4. Lnc Gm16410 Is Involved in the Regulation of Neuronal Necroptosis in AD Under PM_2.5_ Exposure via p38 MAPK Pathway

The p38 MAPK pathway is involved in the regulation of various cellular functions and plays an important role in the occurrence and progression of various diseases [44,45]. To explore whether the p38 MAPK pathway plays a role in PM_2.5_-mediated AD progression, a Western blot assay was employed to detect the levels of p38 and phosphorylated p38 (p-p38) protein in HT22 and SH-SY5Y cells exposed to PM_2.5_. Our findings revealed that the phosphorylation level of p38 protein was elevated in the Aβ_25-35_ and Aβ_25-35_+PM_2.5_ groups compared to the control group. Furthermore, as exposure time lengthened, the phosphorylation level of p38 protein in the Aβ_25-35_+PM_2.5_ group increased progressively, with the most pronounced change observed at 48 h. The phosphorylation level of p38 protein in the Aβ_25-35_+PM_2.5_ 48 h group was significantly higher than that in the Aβ_25-35_ group. (Figure 3A,B). These results indicate that PM_2.5_ exposure promotes the activation of the p38 MAPK pathway in AD neurons in a time-dependent manner. To further investigate the association between the Lnc Gm16410 and p38 MAPK pathways with PM_2.5_-mediated AD, cells in the control group and the OE-Lnc Gm16410 group were exposed to Aβ_25-35_ and PM_2.5_. Western blot analysis revealed that in HT22 cells, the phosphorylation level of p38 protein was elevated in the Aβ_25-35_+PM_2.5_ group compared to the Aβ_25-35_ group, while the expression of p-p38 protein was significantly decreased following the overexpression of Lnc Gm16410 (Figure 3C). Subsequently, SB203580, an inhibitor of the p38 MAPK pathway, was utilized. A Western blot assay was used to detect the changes in necroptosis-related indexes in HT22 cells. It was observed that the levels of RIPK1 and MLKL protein phosphorylation were increased in the Aβ_25-35_+PM_2.5_ group compared to the Aβ_25-35_ group in HT22 cells, and these levels were reversed upon treatment with SB203580 (Figure 3D). Then, the alterations in AD-related markers within HT22 cells were assessed using a Western blot assay. It was observed that the expression of IL-1β and TNF-α protein was elevated in the Aβ_25-35_+PM_2.5_ group compared to the Aβ_25-35_ group, Conversely, the expression of IL-1β and TNF-α protein was significantly decreased following the administration of SB203580 (Figure 3E). The qRT-PCR experiment also revealed the same trend. (Figure 3F). These results indicate that Lnc Gm16410 is involved in the regulation of neuronal necroptosis in AD via the p38 MAPK pathway under PM_2.5_ exposure.

### 2.5. SVHRSP Alleviates Neuronal Necroptosis in AD Under PM_2.5_ Exposure by Lnc Gm16410

Studies have shown that SVHRSP has a variety of neuroprotective effects [39,40,41]. To explore the mechanism and effect of SVHRSP on AD neuron cells exposed to PM_2.5_, a CCK8 test was conducted, and it was found that the survival rate of HT22 neuron cells exhibited minimal change with the increase in SVHRSP treatment concentration (Figure 4A), indicating that SVHRSP showed no neurotoxicity within the appropriate concentration range. Based on preliminary laboratory research [46], the subsequent in vitro medicinal concentration used was 20 μg/mL. To further verify whether SVHRSP plays a role in PM_2.5_-mediated AD, the Aβ_25-35_ and PM_2.5_ co-exposed groups were treated with SVHRSP. The qRT-PCR results indicated that the mRNA expression of Lnc Gm16410 in the Aβ_25-35_+PM_2.5_ group decreased in HT22 cells compared to that in the Aβ_25-35_ group, and this situation was improved following SVHRSP treatment (Figure 4B). Next, the changes in necroptosis-related indexes in HT22 cells were detected by a Western blot assay. It was found that the protein phosphorylation levels of RIPK1 and MLKL in the Aβ_25-35_+PM_2.5_ group were increased compared to those in the Aβ_25-35_ group, while the protein phosphorylation levels of RIPK1 and MLKL were decreased after SVHRSP treatment (Figure 4C). Then, a Western blot and qRT-PCR were utilized to detect alterations in AD-related markers within HT22 cells. It was observed that the levels of IL-1β, TNF-α protein, and mRNA in the Aβ_25-35_+PM_2.5_ group were elevated in the Aβ_25-35_+PM_2.5_ group compared to the Aβ_25-35_ group, while the levels of IL-1β and TNF-α protein and mRNA were notably reduced after SVHRSP treatment (Figure 4D,E). These findings indicated that SVHRSP alleviated the necroptosis of AD neuronal cells under PM_2.5_ exposure through Lnc Gm16410.

### 2.6. SVHRSP Alleviates Cognitive Impairment in AD Mice Exposed to PM_2.5_

To further investigate the impact of PM_2.5_ on AD and the reparative effect of SVHRSP, an AD mouse model was established by injecting Aβ_25-35_ protein into the hippocampus of mice using a brain stereotaxator instrument. Subsequently, the mice were exposed to PM_2.5_ and treated with SVHRSP (Figure 5A). In order to further verify the alterations in mouse brain tissue, HE and Nissl staining were performed on mouse brain tissue. It was observed that the cells in the CA1 region of the hippocampus from the control, PM_2.5,_ and SVHRSP groups were organized into thick, dense, and orderly layers, with intact nuclei. In contrast, the cell layer in the hippocampal CA1 region of the AD and AD+PM_2.5_ groups was thin and indistinct, the cell arrangement was loose and chaotic, and a substantial number of neurons were missing. Compared to the AD group, the AD+PM_2.5_ group exhibited more severe neuron loss and disorganization. However, these pathological changes were mitigated in the AD+PM_2.5_+SVHRSP group (Figure 5B,C). Although no significant pathological alterations were noted in the hippocampus of the PM_2.5_ group compared to the control group, HE staining of the lung tissue revealed thickening of the alveolar wall, structural disarray, and obvious inflammatory cell infiltration in the PM_2.5_ group (Figure 5D).

The Morris water maze experiment was utilized to evaluate changes in cognitive function. It was observed that the escape latency of mice in the AD+PM_2.5_ group was higher than that in the AD group on day 5, and this latency decreased following treatment with SVHRSP. There was no significant difference in average speed among the aforementioned groups (Figure 5E). Throughout the initial 5 days of the Morris water maze assay, the escape latency of mice in all groups decreased, with the AD+PM_2.5_ group showing a lesser decrease compared to the AD group. This was improved after SVHRSP treatment (Figure 5F). During the probe trial, the number of platform crossings in the AD+PM_2.5_ group was lower than that in both the AD group and AD+PM_2.5_+SVHRSP group, although this difference lacked statistical significance (Figure 5G). These outcomes indicate that SVHRSP can alleviate the exacerbation of cognitive impairment in AD mice resulting from PM_2.5_ exposure.

### 2.7. SVHRSP Regulates the Expression of Lnc Gm16410 and Necroptosis in AD Mice Exposed to PM_2.5_

To further verify the effects of PM_2.5_ exposure on AD mice and the role of SVHRSP, ELISA kits were utilized to measure serum levels of IL-1β and TNF-α in the mice. The findings indicated that the levels of IL-1β and TNF-α were elevated in the AD group compared to the control group, and these levels were further elevated following exposure to PM_2.5_. However, this condition was ameliorated after SVHRSP treatment (Figure 6A). The levels of Lnc Gm16410 in the hippocampus were detected using qRT-PCR. The results indicated that the mRNA expression of Lnc Gm16410 in mice from the AD group was reduced compared to the control group. Furthermore, the mRNA expression of Lnc Gm16410 in mice from the AD+PM_2.5_ group was lower than that in the AD group. This condition was ameliorated following SVHRSP treatment (Figure 6B). Subsequently, a Western blot assay was employed to assess the alterations in necroptosis and the related indexes of the p38 MAPK pathway within the brain tissue of mice hippocampi. The findings revealed that the protein phosphorylation levels of RIPK1, MLKL, and p38 in the AD group were elevated compared to the control group. These levels were further heightened after exposure to PM_2.5_, yet the situation was improved after SVHRSP treatment (Figure 6C,D). Then, a Western blot and qRT-PCR were utilized to detect alterations in inflammation-related indexes within the hippocampus brain tissue of mice. The findings indicated that the expression of IL-1β, TNF-α protein, and mRNA in the AD group was elevated compared to the control group, with the levels being exacerbated upon exposure to PM_2.5_. These changes were ameliorated following treatment with SVHRSP (Figure 6E,F).

## 3. Discussion

Air pollution is one of the major environmental risk factors to global human health. PM_2.5_, as one of the main pollutants in the air, can cause great harm to the human body because of its small particle size and toxic and harmful substances. AD is the leading cause of dementia and is quickly becoming one of the most expensive, deadly, and burdensome diseases of this century [47]. Epidemiological studies have demonstrated that prolonged exposure to PM_2.5_ air pollution exacerbates the risk of AD and dementia [48]. This may be due to the ability of PM_2.5_ to enter the blood circulation, disrupt the blood–brain barrier, and cause neuroinflammation [49,50]. PM_2.5_ can also bypass the blood–brain barrier via olfactory nerve transport and affect the central nervous system [51]. However, whether necroptosis plays a role in PM_2.5_-mediated AD has not been further elucidated. In this study, we constructed in vivo and in vitro AD models to investigate the effect of PM_2.5_ exposure on the progression of necroptosis in the AD model and explore the mitigating effects and underlying mechanisms of SVHRSP.

Necroptosis is a form of programmed cell death mediated by RIPK1, RIPK3, and MLKL that plays an important role in many diseases. Although relevant studies have shown that AD is associated with a variety of cell death modes, such as ferroptosis and pyroptosis [52,53], there are relatively few studies on the relationship between AD and necroptosis. In this experiment, we chose to detect RIPK1, p-RIPK1, MLKL, and p-MLKL to reflect the degree of necroptosis. The CCK8 assay was employed to determine the concentrations of PM_2.5_ and Aβ_25-35_ in HT22 and SH-SY5Y cells. The effects of PM_2.5_ on necroptosis and AD were investigated by constructing an in vitro model. Experiments, including WB, qRT-PCR, and others, have demonstrated that exposure to PM_2.5_ accelerates neuronal cell necroptosis in AD at both the protein and gene levels, with time-dependent effects.

Lnc Gm16410 is a long non-coding RNA that is highly correlated with PM_2.5_ exposure, which was identified by high-throughput sequencing technology in previous laboratory studies. Studies have shown that Lnc Gm16410 is involved in PM_2.5_-induced endothelium–mesenchymal transformation and macrophage activation [35,36]. In this study, we investigated the potential involvement of LncRNA Gm16410 in the progression of AD, particularly focusing on the exacerbation of the condition by PM_2.5_ exposure. Our research indicates that Aβ_25-35_, a peptide fragment implicated in AD, negatively regulates the expression of Lnc Gm16410 in HT22 cells, and this down-regulated trend was more significant after co-exposure with PM_2.5_. Hence, we delved deeper into the underlying mechanism of Lnc Gm16410’s action. The results showed that the overexpression of Lnc Gm16410 could effectively reduce the necroptosis of neuronal cells and the progression of AD induced by PM_2.5_ exposure.

The p38 MAPK pathway can respond to various cellular stimuli mediated by inflammation and aging, and plays a crucial role in many biological processes [54]. In this experiment, we chose to detect p38 and p-p38 to reflect the activation degree of the p38 MAPK pathway. Our study explored the function of the p38 MAPK pathway in the development of AD induced by PM_2.5_ exposure. The experimental findings demonstrated that the activation of the p38 MAPK pathway was enhanced in AD neurons exposed to PM_2.5_, and this enhancement was time-dependent. Subsequently, we administered SB203580, an inhibitor of the p38 MAPK pathway. Following the introduction of SB203580, the necroptosis and inflammatory markers associated with AD in neuronal cells were suppressed in the presence of both Aβ_25-35_ and PM_2.5_. Subsequent research revealed that the p38 MAPK pathway, a crucial component in cellular regulation, experienced significant inhibition upon the overexpression of Lnc Gm16410. Hence, we conclude that Lnc Gm16410 may play a role in the necroptosis of neuronal cells in AD under PM_2.5_ exposure, potentially via the p38 MAPK pathway.

SVHRSP has anti-inflammatory and neuronal protective effects. In this study, to ensure the safety of SVHRSP, we conducted a CCK8 assay to assess its potential neurotoxicity across a range of concentrations; the assay demonstrated that SVHRSP exhibits no neurotoxic effects within the tested concentration range. Following the administration of SB203580, the suppression of Lnc Gm16410 expression under the combined influence of Aβ_25-35_ and PM_2.5_ was alleviated. Additionally, the neuronal necroptosis and inflammatory markers associated with AD were also attenuated. Therefore, we conclude that SVHRSP can alleviate the PM_2.5_-mediated neuronal cell necroptosis and AD progression through Lnc Gm16410. To further verify the relevant mechanisms, we constructed an in vivo model of AD exposed to PM_2.5_ and treated it with SVHRSP via intraperitoneal injection. The Morris water maze experiment revealed that SVHRSP treatment can alleviate the cognitive impairment of AD mice promoted by PM_2.5_ exposure. HE and Nissl staining of brain tissue revealed no apparent pathological alterations in the PM_2.5_ group, whereas HE staining of lung tissue indicated structural changes in the alveolar wall of the PM_2.5_ group; the structure appeared disorganized, with inflammatory cell infiltration clearly evident, indicating that the tracheal injection modeling of PM_2.5_ was successful. When compared to the AD group, the cells in the hippocampal CA1 region of the brain tissue from the AD+PM_2.5_ group displayed a loose and disordered arrangement, accompanied by a significant loss of neurons, suggesting that exposure to PM_2.5_ may exacerbate the pathological changes in AD neurons. The pathological changes observed in brain tissue within the AD+PM_2.5_+SVHRSP group exhibited improvement, further strengthening the evidence for SVHRSP’s capacity to mitigate the progression of AD mediated by PM_2.5_. Furthermore, elevated levels of IL-1β and TNF-α in mouse serum, coupled with the results from WB and qRT-PCR experiments, confirmed that exposure to PM_2.5_ can induce necroptosis in brain tissue, activate the p38 MAPK pathway, and elevate AD-related inflammatory markers in AD mice. Notably, SVHRSP treatment was found to ameliorate these effects.

This study uncovered that Lnc Gm16410 potentially alleviates the process of necroptosis in AD neurons via the p38 MAPK pathway and also explored the repair effects and mechanisms of SVHRSP when exposed to PM_2.5_ (Figure 7), thereby offering fresh theoretical and experimental foundations for the early diagnosis, treatment, and prognosis of AD in PM_2.5_-exposed environments. Nonetheless, the experiment does possess certain limitations; for instance, the subcellular localization of Lnc Gm16410 still awaits further exploration, and the precise mechanism through which Lnc Gm16410 regulates the p38 MAPK pathway has yet to be thoroughly investigated.

## 4. Conclusions

Lnc Gm16410 promotes the progression of AD by regulating neuronal necroptosis via the p38 MAPK pathway under PM_2.5_ exposure. SVHRSP alleviates PM_2.5_-induced necroptosis and cognitive impairment in AD mice by regulating Lnc Gm16410, which is a potential regulator of AD progression.

## 5. Materials and Methods

### 5.1. Chemicals and Reagents

Rabbit antibodies against RIPK1, p-RIPK1 (Ser166), MLKL, and p-MLKL (Ser358) were purchased from Affinity (Cincinnati, OH, USA). Rabbit antibody against p-MLKL (Ser345) was purchased from Bioss (Beijing, China). Rabbit antibodies against p38, p-p38, IL-1β, and TNF-α were purchased from Proteintech (Wuhan, China). IL-1β and TNF-α ELISA kits were purchased from Elabscience Biotechnology (Wuhan, China). The cell counting kit-8 (CCK8) (C0038) was purchased from Beyotime (Shanghai, China). All reverse transcription and real-time PCR kits were purchased from TransGen Biotech (Beijing, China).

### 5.2. Extraction and Preparation of PM_2.5_ Samples

Ultrafine quartz fiber filters was purchased from General Electric (Boston, MA, USA) were used to collect PM_2.5_ in (Dalian, China) from October 2021 to March 2022. The quartz filter paper was cut and weighed in double-distilled water. The sonication process was repeated five times, with each session lasting 20 min, to effectively remove PM_2.5_ particles. The filter paper was dried and weighed, and the difference between its weight before and after sonication was calculated to estimate the concentration of PM_2.5_.

### 5.3. Preparation and Processing of Aβ Protein

Aβ_25-35_ is a white powder with the amino acid sequence N-Gly-Ser-Asn-Lys-Gly- Ala-Ile-Ile-Gly-Leu-Met-C and a relative molecular weight of 1060.27. The Aβ_25-35_ powder was dissolved into a 5 mM stock solution using DMSO and stored in the refrigerator at −80 °C. According to the experimental arrangement, the Aβ_25-35_ working solution was prepared using normal saline in vivo, with a concentration of 2 μg/μL. In in vitro experiments, the Aβ_25-35_ working solution was prepared using simple medium according to the required concentration.

### 5.4. Preparation of SVHRSP

SVHRSP was provided by the National and Local Joint Engineering Research Center for Drug Development of Neurodegenerative Diseases, Dalian Medical University. Its amino acid sequence is N-Lys-Val-Leu-Asn-Gly-Pro-Glu-Glu-Glu-Ala-Ala-Ala-Pro- Ala-Glu-C and its molecular weight is 1524.66. According to the experimental arrangement, normal saline was used to prepare the SVHRSP working liquid in vivo. In vitro experiment, the SVHRSP working liquid was prepared using a simple medium.

### 5.5. Animal Experiments and Experiment Design

All procedures involving animals were approved by the Animal Experiment Committee of Dalian Medical University (AEE21058). A total of 36 C57BL/6J mice (aged 9–13 weeks) were procured from Liaoning Changsheng Biotechnology Co., Ltd. (Shenyang, China) and housed in cages under specific pathogen-free (SPF) conditions on a 12/12 h light/dark cycle at 20–22 °C.

The mice were randomly divided into six groups, each comprising 6 mice, as follows: control, AD, PM_2.5_, SVHRSP, AD+PM_2.5,_ and AD+PM_2.5_+SVHRSP groups. Initially, mice in the AD group, AD+PM_2.5_ group, and AD+PM_2.5_+SVHRSP group were injected with Aβ_25-35_ protein (2 μg/μL) into the hippocampus using a brain stereotaxic instrument, while the remaining groups received an equal volume of normal saline. Next, mice in the PM_2.5_ group, AD+PM_2.5_ group, and AD+PM_2.5_+SVHRSP group were injected with PM_2.5_ (10 mg/μL, 50 μL, every 2 days, for a total of 4 times), while the remaining groups received an equal volume of normal saline. Subsequently, the SVHRSP group and AD+PM_2.5_+SVHRSP group were intraperitoneally injected with SVHRSP (200 μg/kg, once a day for 23 days), with the other groups receiving an equal volume of normal saline.

### 5.6. Cell Culture

HT22 and SH-SY5Y cells were cultured in DMEM (89% DMEM) supplemented with 10% fetal bovine serum and 1% double antibiotics and maintained in a humidified incubator with 5% CO₂ at 37 °C for 2–3 days. The cells were passaged when they reached 80–90% confluency.

### 5.7. CCK8

The groups were divided into a control group, an experimental group, and a zero setting group, each with four multiple holes. A cell suspension of 100 μL was successively added to each well of the 96-well cell culture plate, resulting in approximately 2 × 10^4^ cells per well. Once the cells were fully attached to the walls, different concentrations of PM_2.5_, Aβ_25-35_, or SVHRSP were introduced according to the experimental groups. Following the treatment period, the drug was removed, and 100 μL of DMEM medium and 10 μL of CCK8 reagent were added to each well. The culture was then continued for an additional 2 h. Subsequently, the 96-well plate was removed, and the absorbance value of each well at 450 nm was measured using an enzyme labeling instrument, followed by statistical analysis.

### 5.8. Western Blot

Refer to Appendix A for details.

### 5.9. RNA Extraction and Real-Time PCR

The sequences of all primers used and details are listed in Appendix A.

### 5.10. Morris Water Maze

The water maze consists of a circular pool, a movable platform, and a video tracking system. The experiment was divided into two phases: the spatial navigation (the first 5 days) and the probe trials (on the sixth day), which together lasted for a total of 6 days. The pool is divided into four quadrants (I, II, III, IV), and the platform is placed in the center of the third quadrant, with its bottom 2 cm below the liquid level. The localization cruise experiment lasted for 5 days, and the time from entering the water to finding the platform was the latency escape period. If the tablet was not located within the prescribed 60 s, the mice were artificially guided to stand on the tablet for 10 s, and the latency escape period was recorded as 60 s. On the 6th day, the platform withdrawal experiment was carried out, the plate was removed, and the frequency of the mice entering the water and crossing over the plate was recorded.

### 5.11. HE Staining

The prepared paraffin sections were dewaxed and hydrated with xylene. Hematoxylin staining was used, differentiation liquid was differentiated, blue returning liquid was returned to blue, and distilled water was washed. The slices were dehydrated in anhydrous ethanol for 1 min and subsequently dyed with eosin solution for 2 min. Each slice was dehydrated using anhydrous ethanol, followed by xylene. Neutral gum was used for sealing. Tissue images were observed and captured under an optical microscope.

### 5.12. Nissl Staining

The paraffin sections were dewaxed and hydrated. The tissue was then soaked in Nissl dye for 2–5 min and rinsed with distilled water. Then, 0.1% glacial acetic acid was differentiated and rinsed with distilled water. Images of brain tissue in the hippocampus were observed and captured under an optical microscope.

### 5.13. ELISA

The standard product and the sample to be tested were added into the enzyme label plate successively, with 50 μL per well, and incubated at 37 °C for 60 min. Discard the liquid, add 100 μL of biotin-labeled antibody per well, and incubate at 37 °C for 30 min. Discard the liquid, soak the wells with washing liquid for 2 min, discard the liquid from the wells, pat dry on filter paper, and repeat this process three times. Add 50 μL of each of the two color developing agents to each well and incubate at 37 °C for 15 min in the absence of light. Finally, add 50 μL of termination solution to each well. The OD values of each well were detected at a 450 nm wavelength using Origin software to fit standard curves, and the contents of TNF-α and IL-1β in each sample were calculated.

### 5.14. Statistical Analysis

Data are presented as means ± SEMs. Statistical analyses were performed using GraphPad Prism 8 (GraphPad, San Diego, CA, USA). Student’s *t*-tests and one-way ANOVA were employed according to actual conditions. A value of *p* < 0.05 was regarded as statistically significant.

## Figures and Tables

**Figure 1 ijms-26-04372-f001:**
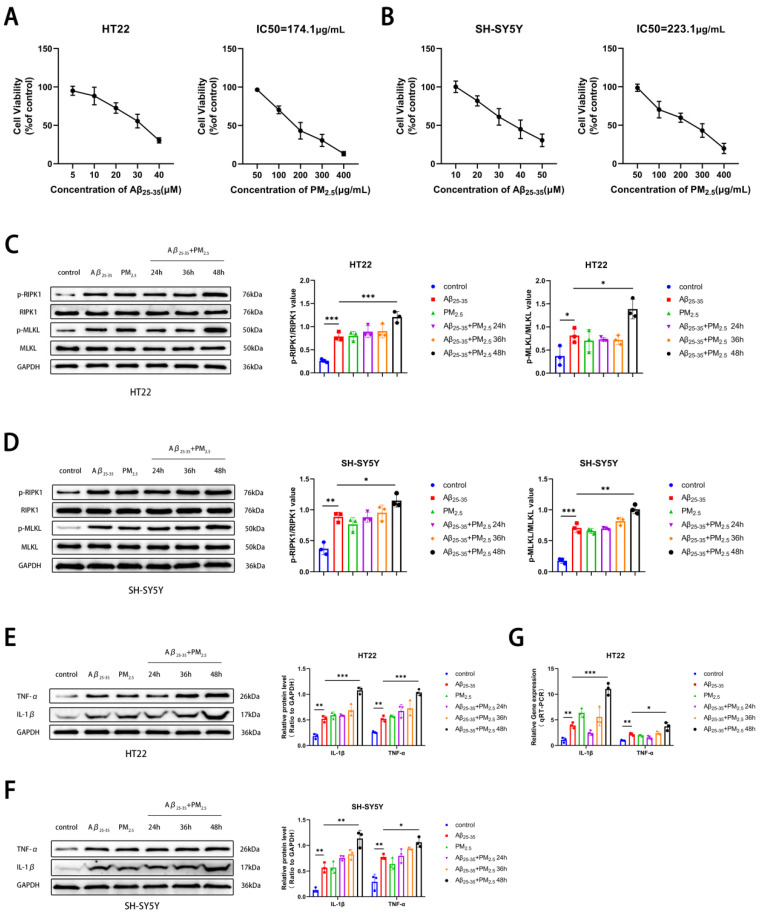
PM_2.5_ exposure promotes the necroptosis of AD neuronal cells. (**A**) Survival rate changes in HT22 cells exposed to varying concentrations of Aβ_25-35_ or PM_2.5_ (*n* = 3). (**B**) Survival rate changes in SH-SY5Y cells exposed to varying concentrations of Aβ_25-35_ or PM_2.5_ (*n* = 3). (**C**) Expression of RIPK1, p-RIPK1, MLKL, and p-MLKL in HT22 cells and analysis of their gray values (*n* = 3). (**D**) Expression of RIPK1, p-RIPK1, MLKL, and p-MLKL in SH-SY5Y cells and analysis of their gray values (*n* = 3). (**E**) Expression of IL-1β and TNF-α proteins in HT22 cells and analysis of their gray values (*n* = 3). (**F**) Expression of IL-1β and TNF-α proteins in SH-SY5Y cells and analysis of their gray values (*n* = 3). (**G**) mRNA levels of IL-1β and TNF-α in HT22 cells (*n* = 3). (“*”, “**”, and “***” means *p* < 0.05, *p* < 0.01, and *p* < 0.001, respectively).

**Figure 2 ijms-26-04372-f002:**
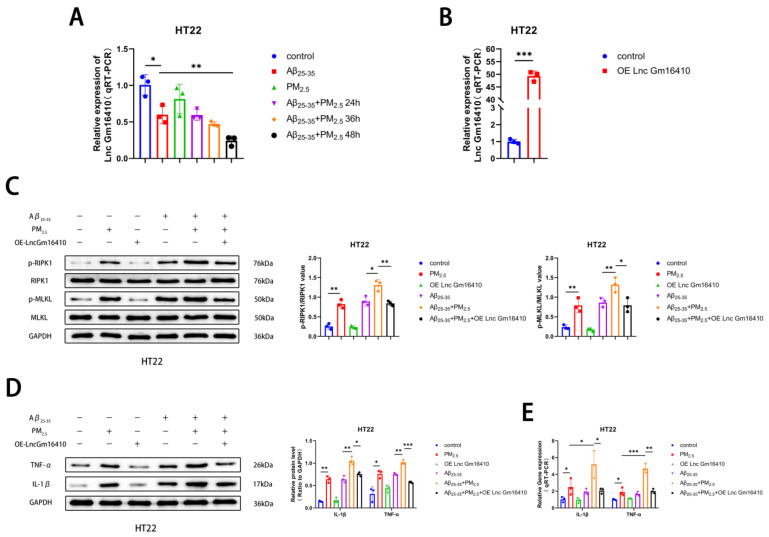
Lnc Gm16410 plays a role in the regulation of neuronal necroptosis in AD following exposure to PM_2.5_. (**A**) The mRNA levels of Lnc Gm16410 in HT22 cells (*n* = 3). (**B**) Transfection efficiency of Lnc Gm16410 in HT22 cells (*n* = 3). (**C**) Expression of RIPK1, p-RIPK1, MLKL, and p-MLKL proteins and gray value analysis after overexpression of Lnc Gm16410 in HT22 cells (*n* = 3). (**D**) Expression and gray value analysis of IL-1β and TNF-α proteins after overexpression of Lnc Gm16410 in HT22 cells (*n* = 3). (**E**) The mRNA levels of IL-1β and TNF-α after overexpression of Lnc Gm16410 in HT22 cells (*n* = 3). (“*”, “**”, and “***” means *p* < 0.05, *p* < 0.01, and *p* < 0.001, respectively).

**Figure 3 ijms-26-04372-f003:**
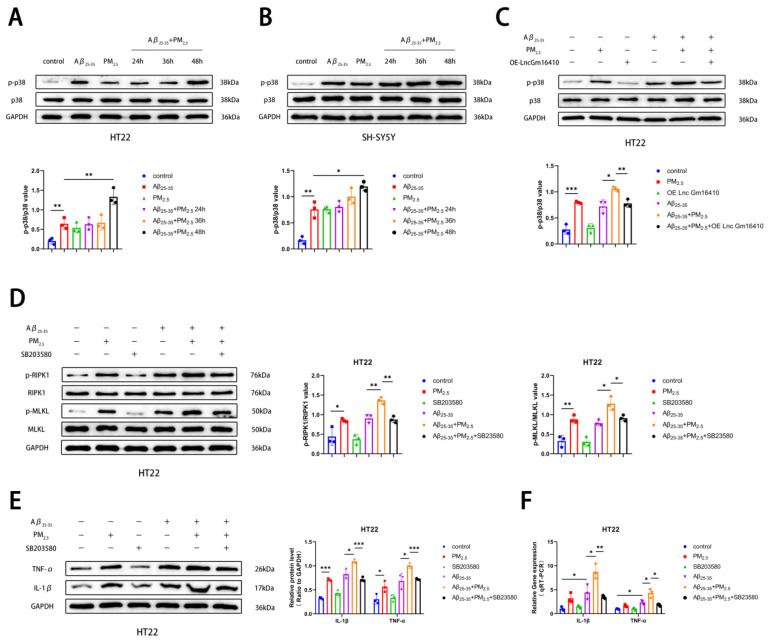
Lnc Gm16410 is involved in the regulation of neuronal necroptosis in AD under PM_2.5_ exposure via p38 MAPK pathway. (**A**) Expression of p38 and p-p38 proteins in HT22 cells and gray value analysis (*n* = 3). (**B**) Expression of p38 and p-p38 proteins in SH-SY5Y cells and analysis of their gray values (*n* = 3). (**C**) Expression and gray value analysis of p38 and p-p38 proteins in HT22 cells treated with SB203580 (*n* = 3). (**D**) Expression and gray value analysis of RIPK1, p-RIPK1, MLKL, and p-MLKL proteins in HT22 cells treated with SB203580 (*n* = 3). (**E**) Expression and grayscale analysis of IL-1β and TNF-α proteins in HT22 cells treated with SB203580 (*n* = 3). (**F**) The mRNA levels of IL-1β and TNF-α in HT22 cells treated with SB203580 (*n* = 3). (“*”, “**”, and “***” means *p* < 0.05, *p* < 0.01, and *p* < 0.001, respectively).

**Figure 4 ijms-26-04372-f004:**
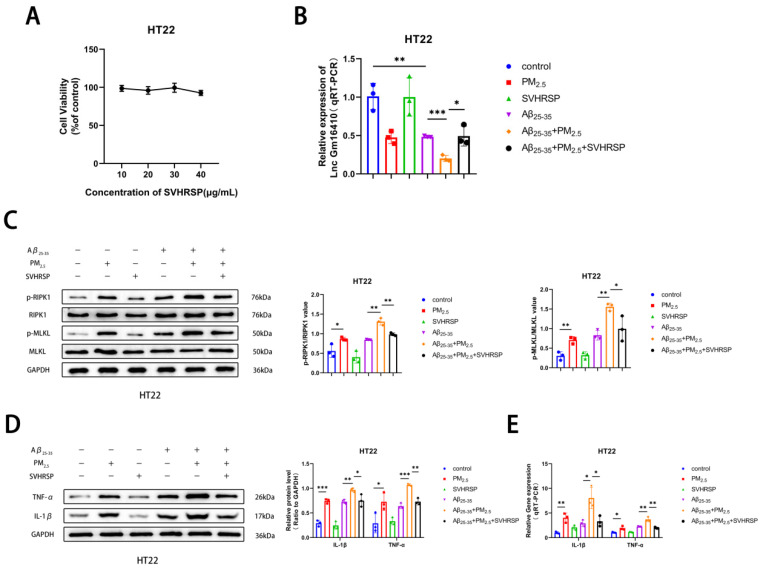
SVHRSP alleviates neuronal necroptosis in AD under PM_2.5_ exposure by Lnc Gm16410. (**A**) The CCK8 assay was utilized to determine the survival rate of HT22 cells treated with varying concentrations of SVHRSP (*n* = 3). (**B**) The mRNA levels of Lnc Gm16410 in HT22 cells treated with SVHRSP were detected by qRT-PCR assay (*n* = 3). (**C**) The Western blot assay was utilized to detect the expression of RIPK1, p-RIPK1, MLKL, and p-MLKL proteins in HT22 cells following treatment with SVHRSP, and their gray values were subsequently analyzed (*n* = 3). (**D**) The Western blot assay was utilized to detect the expression of IL-1β and TNF-α proteins in HT22 cells treated with SVHRSP, and analyze their gray values (*n* = 3). (**E**) The mRNA levels of IL-1β and TNF-α in HT22 cells treated with SB203580 were detected by a qRT-PCR assay (*n* = 3). (“*”, “**”, and “***” means *p* < 0.05, *p* < 0.01, and *p* < 0.001, respectively).

**Figure 5 ijms-26-04372-f005:**
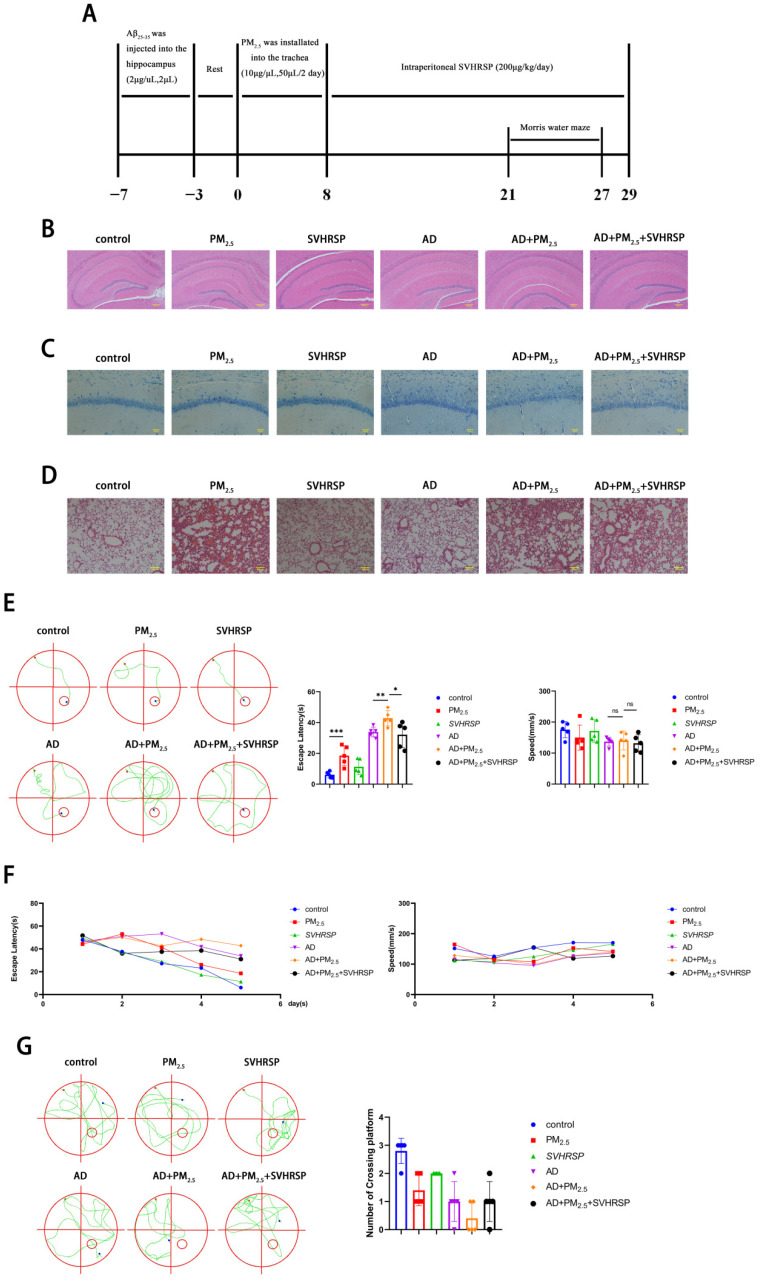
SVHRSP alleviates cognitive impairment in AD mice exposed to PM_2.5_. (**A**) Timeline of in vivo experiments. (**B**) HE staining in the hippocampus of mouse brain tissue (scale = 200 μm). (**C**) Nissl staining of hippocampus of mouse brain tissue (scale = 50 μm). (**D**) HE staining of mouse lung tissue (scale = 100 μm). (**E**) Analysis of track chart, escape latency, and average speed on day 5 of the water maze positioning sailing experiment (*n* = 5). (**F**) Analysis of escape latency and average speed during the first 5 days of the water maze positioning navigation experiment (*n* = 5). (**G**) Trajectory chart and analysis of the number of platform crossings on day 6 of the Morris water maze experiment (*n* = 5). (“*”, and “**”, and “***” means *p* < 0.05, *p* < 0.01, and *p* < 0.001, respectively; “ns” means non-significant difference).

**Figure 6 ijms-26-04372-f006:**
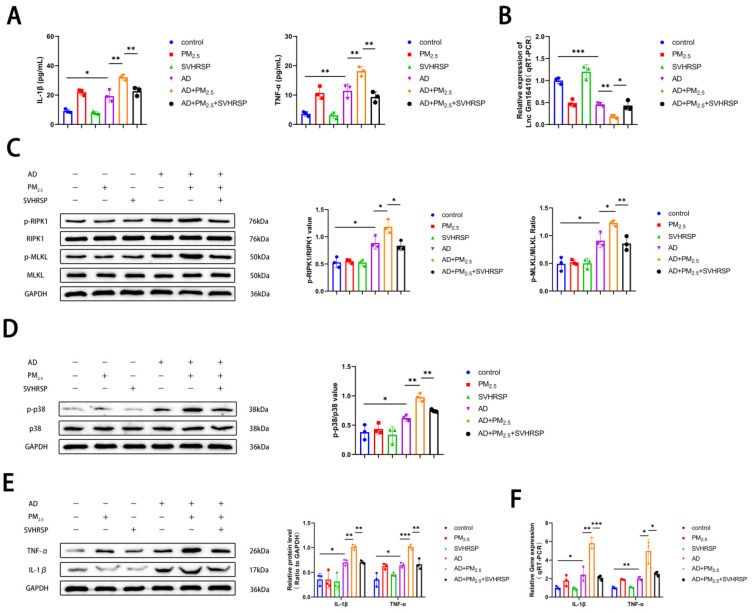
SVHRSP regulates the expression of Lnc Gm16410 and necroptosis in AD mice exposed to PM_2.5_. (**A**) Serum levels of IL-1β and TNF-α in mice (*n* = 3). (**B**) mRNA level of Lnc Gm16410 in hippocampus of mouse brain tissue (*n* = 3). (**C**) Expression of RIPK1, p-RIPK1, MLKL, and p-MLKL proteins in hippocampus of mouse brain tissue and analysis of their gray values (*n* = 3). (**D**) Expression of p38 and p-p38 proteins in hippocampus of mouse brain tissue and analysis of their gray values (*n* = 3). (**E**) Expression and gray value analysis of IL-1β and TNF-α proteins in hippocampus of mouse brain tissue (*n* = 3). (**F**) mRNA levels of IL-1β and TNF-α in hippocampus of mouse brain tissue (*n* = 3). (“*”, “**”, and “***” means *p* < 0.05, *p* < 0.01, and *p* < 0.001, respectively).

**Figure 7 ijms-26-04372-f007:**
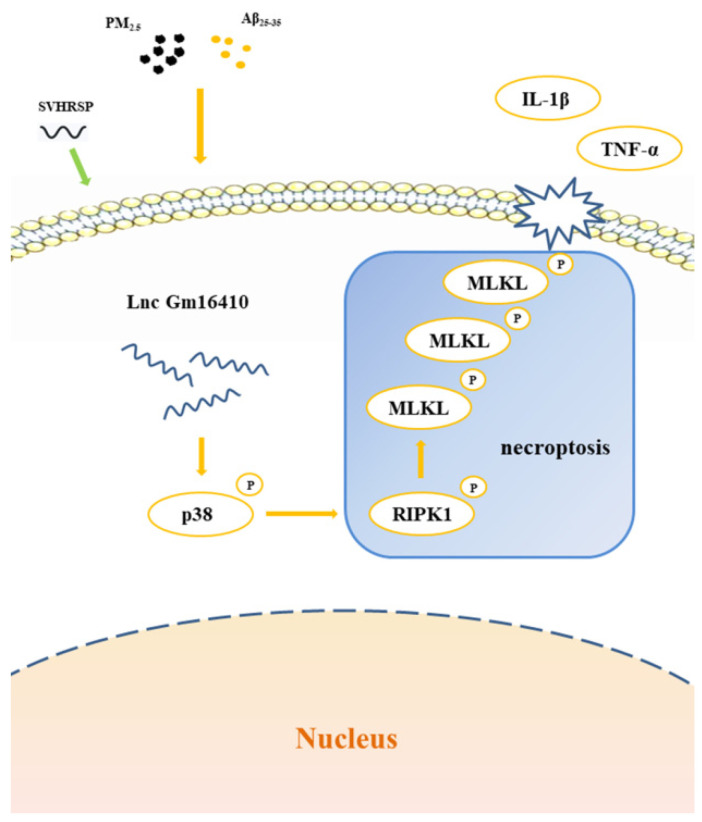
SVHRSP alleviates neuronal necroptosis in the AD model by regulating Lnc Gm6410 under PM_2.5_ exposure.

## Data Availability

Data will be made available upon request.

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
