# Peer review of "Scorpion Venom Heat-Resistant Synthetic Peptide Alleviates Neuronal Necroptosis in Alzheimer’s Disease Model by Regulating Lnc Gm6410 Under PM2.5 Exposure"

_ijms, 2025, doi:10.3390/ijms26094372_

Round 1
Reviewer 1 Report
Comments and Suggestions for Authors
In this study the authors investigated how SVHRSP alleviates neuronal necroptosis in the Alzheimer’s disease model by regulating Lnc GM6410 under PM2.5 exposure. The experimental design and results presented supported authors conclusion. The study has potential for high impact. However, some major concerns for the authors to consider
- Line 25 The statement “Furthermore, they propose promising strategies for drug therapy of AD by investigating the mechanism of action of SVHRSP in mitigating AD” seem out of place should be check or reframe.
- 5 is large can not enter the circulatory system, reside in the lung, how can this cross the blood brain barrier?
- All the figures’ legends do not have numbers of trials (replications). Can the authors put down the number of N for each study in their respective figure legends. How many independent trials where use for each experiment.
- Spacing issue throughout the manuscript
- Figure 5 B, C & D need quantification
- Conclusion should be in the form of paragraph comprising few sentences not 2 bullet points. I suggest summarizing the key findings or the take home message in a paragraph.
Author Response
- Line 25 The statement “Furthermore, they propose promising strategies for drug therapy of AD by investigating the mechanism of action of SVHRSP in mitigating AD” seem out of place should be check or reframe.
Response: We apologize for the error in the sentence. Following your suggestion, we have revised the sentence in the abstract to read: “Furthermore, we offer new targets for the treatment and prevention of AD following PM2.5 exposure by investigating the mechanism of action of SVHRSP in alleviating AD.”(Page 1, line 25-27)
- 2.5 is large can not enter the circulatory system, reside in the lung, how can this cross the blood brain barrier?
Response: Thanks for your suggestion regarding this article. PM2.5 refers to particulate matter with an aerodynamic diameter of 2.5 μm or less. The PM2.5 components analysed include organic carbon, elemental carbon, water-soluble ions (such as NH4+, NO3-, SO42-, etc.), metallic elements (including Al, Ca, Fe, etc.) and polycyclic aromatic hydrocarbons (like naphthalene, acenaphthene, fluoranthene, etc.), respectively. Relevant literature indicates that most PM2.5 (approximately 95%) is inhaled from the air through the mouth and nose and then passes through the blood barrier of the lungs, while the remainder (approximately 5%) is absorbed via the gastrointestinal tract. Biological experiments demonstrate that PM2.5 can enter the brain by disrupting the tight junctions of the blood-brain barrier. Additionally, PM2.5 can also enter the gastrointestinal tract, causing imbalances in the intestinal microecology that may affect central nervous system diseases . Consequently, PM2.5 exposure has a certain theoretical basis for brain damage. We have added relevant literature in the discussion section of the revised draft for supplementary explanations.(Page 11, line 370-373)
- All the figures’ legends do not have numbers of trials (replications). Can the authors put down the number of N for each study in their respective figure legends. How many independent trials where use for each experiment.
Response: Thank you very much for your suggestions on this article. Following your suggestion, we have added the number of experimental replicates to the figure legend in the revised manuscript.
- Spacing issue throughout the manuscript
Response: Thank you very much for your comments on this article. Following your suggestion, we have carefully revised the layout and format of the manuscript.
- Figure 5 B, C & D need quantification
Response: We sincerely appreciate the reviewer's thoughtful suggestion regarding the quantitative analysis of HE and Nissl staining. While we acknowledge the potential value of quantitative approaches in histopathological studies, our primary aim is to demonstrate pathological damage to tissues , such as neuronal disorganization. These observations are typically qualitative and focus on morphological changes. Consequently, we have added directional arrows in Figures 5B, 5C, and 5D to facilitate a clearer observation of the relevant pathological changes.(Page 10)
- Conclusion should be in the form of paragraph comprising few sentences not 2 bullet points. I suggest summarizing the key findings or the take home message in a paragraph.
Response: Thank you very much for your comments on this article. Following your suggestion, we revised the conclusion section and summarized the main findings in a single paragraph.(Page 14, line 466-469)
Reviewer 2 Report
Comments and Suggestions for Authors
Manuscript #:ijms-3579399 Chuhao Qin et al.
SVHRSP alleviates neuronal necroptosis in the Alzheimer's disease model by regulating Lnc Gm6410 under PM2.5 exposure
The objective of the present study is to determine whether long non-coding RNA Gm16410 and neuronal necroptosis are involved in pathogenesis of Alzheimer's disease (AD) stimulated by PM2.5 exposure, as well as in the mechanisms of scorpion venom SVHRSP in alleviating this process. The results showed that Lnc Gm16410 regulates neuronal necroptosis under PM2.5 exposure via p38 MAPK pathway and SVHRSP is a potential regulator of AD progression by regulating Lnc Gm16410 to alleviate PM2.5 exposure induced necroptosis.
I agree that the paper is interesting and important since these findings offer new insights into the mechanism through which PM2.5 exposure accelerates the progression of AD and may potentially give a clue for the therapy strategy against the PM2.5-induced AD. My comments are 1. As far as I noticed, the data relevant to necroptosis in ‘% cell viability’. Are there electron-microgram of AD brain? Alternatively, are there molecular markers specific to necroptosis available? 2. In many experiments, positive or negative controls may be required (e.g. Control peptide against SVHRSP, Lnc other than Gm16410, A pathway other than p38 MAPK, ,,,)
Author Response
- As far as I noticed, the data relevant to necroptosis in ‘% cell viability’. Are there electron-microgram of AD brain? Alternatively, are there molecular markers specific to necroptosis available?
Response: Thank you very much for your comments on this article. The stepwise phosphorylation of RIPK1 and MLKL is an important mechanism of necroptosis. The levels of p-RIPK1, RIPK1, p-MLKL and MLKL in mouse brain tissues and cell lines were detected by reviewing relevant literature and considering the mechanism of necroptosis. The phosphorylation of RIPK1 and MLKL can reflect the progression of necroptosis. Consequently, we did not perform the electron microscope, and we hope for your understanding.
- In many experiments, positive or negative controls may be required (e.g. Control peptide against SVHRSP, Lnc other than Gm16410, A pathway other than p38 MAPK, ,,,)
Response: We sincerely appreciate the reviewer's insightful suggestion regarding the inclusion of additional positive controls , such as other LncRNA, pathway, or peptide). We fully acknowledge that comparative controls could theoretically strengthen the mechanistic validation. In the past decades, most clinical drugs have been discontinued due to limited effectiveness or adverse effects. Currently, available drugs primarily offer symptomatic relief and are often accompanied by undesirable side effects. The recent approvals of aducanumab and lecanemab by the Food and Drug Administration present the potential in disrease-modifying effects. Nevertheless, the long-term efficacy and safety of these drugs require further validation. Consequently, the quest for safer and more effective AD drugs continues as a formidable and pressing task.
Scorpions have been utilized in Chinese medicine for more than 1,000 years and are extensively employed in traditional Chinese medicine to treat various ailments, including epilepsy, convulsions and rheumatism. Scorpion venom (SV), a principal active component of scorpions, has surfaced as a significant therapeutic agent. Scorpion venom heat resistant peptide (SVHRP) is a type of active polypeptide, extracted from scorpion venom and analyzed for its amino acid sequence using LC-MS. Scorpion venom heat resistant synthetic peptide (SVHRSP) is an active polypeptide synthesized according to its amino acid sequence. The substance boasts properties of low toxicity, high purity, and heat stability, which are beneficial for its application in medical research. Previous studies have indicated that SVHRSP can alleviate neuroinflammation and oxidative stress in vitro. We aimed to investigate whether SVHRSP could impede the progression of AD under PM2.5 exposure and uncover the related mechanism. Consequently, we selected SVHRSP as the primary subject of our study. In this regard, we have provided additional explanations in the intriduction.(Page 3, line 121-126)
Lnc Gm16410 was identified through high-throughput sequencing technology in previous laboratory studies. Research has indicated that Lnc Gm16410 plays a role in PM2.5-induced endothelial-mesenchymal transition and macrophage activation. For the mechanistic investigation, we conducted qRT-PCR on mouse brain tissues and cell lines. Given the significant differences in Lnc Gm16410 expression, we selected it as the primary focus of our research.
The p38 MAPK pathway responds to various cellular stimuli, including those mediated by inflammation and aging, and plays a crucial role in numerous biological processes. Consequently, we selected the p38 MAPK pathway as the focus of our research to explore the relationship between Lnc Gm16410 and necroptosis by knocking down Lnc Gm16410 and utilizing p38 MAPK pathway inhibitors.
Round 2
Reviewer 1 Report
Comments and Suggestions for Authors
I do not have any concern on the manuscript.
Author Response
We deeply appreciate the time and effort you dedicated to evaluating our research, and we are sincerely grateful for your insightful suggestions, which have significantly strengthened the clarity and rigor of the manuscript.
Reviewer 2 Report
Comments and Suggestions for Authors
I am agree that the authors well responded to my comments.
Author Response

(The authors gave the same response as above.)
